

# Mapping sugarcane globally at 10 m resolution using GEDI and Sentinel-2

Stefania Di Tommaso[1], Sherrie Wang[2], Rob Strey[3], and David B. Lobell[1]

[1]Department of Earth System Science & Center on Food Security and the Environment, Stanford University, USA
[2]Department of Mechanical Engineering & Institute for Data, Systems, and Society, MIT, USA
[3]Progressive Environmental & Agricultural Technologies, 10435 Berlin, Germany

**Correspondence:** David B. Lobell (dlobell@stanford.edu)

**Abstract.** Sugarcane is an important source of food, biofuel, and farmer income in many countries. At the same time, sugarcane
is implicated in many social and environmental challenges, including water scarcity and nutrient pollution. Currently, few of the
top sugar-producing countries generate reliable maps of where sugarcane is cultivated. To fill this gap, we introduce a dataset
of detailed sugarcane maps for the top 13 producing countries in the world, comprising nearly 90% of global production. Maps
were generated for the 2019-2022 period by combining data from the Global Ecosystem Dynamics Investigation (GEDI) and
Sentinel-2 (S2). GEDI data were used to provide training data on where tall and short crops were growing each month, while S2
features were used to map tall crops for all cropland pixels each month. Sugarcane was then identified by leveraging the fact that
sugar is typically the only tall crop growing for a substantial fraction of time during the study period. Comparisons with field
data, pre-existing maps, and official government statistics all indicated high precision and recall of our maps. Agreement with
field data at the pixel level exceeded 80% in most countries, and sub-national sugarcane areas from our maps were consistent
with government statistics. Exceptions appeared mainly due to problems in underlying cropland masks, or to under-reporting
of sugarcane area by governments. The final maps should be useful in studying the various impacts of sugarcane cultivation
and producing maps of related outcomes such as sugarcane yields.

## 1  Introduction

Sugarcane cultivation represents an important economic activity in many regions of the world, and serves as a substantial source
of food, beverage, and biofuel production. Roughly one-quarter of all ethanol production worldwide comes from sugarcane
(OECD et al., 2023), with many countries aiming to rapidly increase sugar ethanol production to meet energy independence
and climate mitigation goals. For example, the OECD/FAO projects that ethanol demand over the next decade will increase by
37% in Brazil and 107% in India (OECD et al., 2023), both countries where sugarcane is the primary feedstock. Moreover,
millions of livelihoods are derived from sugarcane production and processing activities, with some estimates putting the total
number of livelihoods dependent on sugarcane as high as 100 million (Jenkins et al., 2015).
Despite its contribution to food and energy security and economic growth, sugarcane cultivation has also been associated
with myriad challenges, including but not limited to large consumption of available freshwater and fertile cropland (Lee et al.,
2020), pollution of soils and ecosystems with nutrients and other chemical runoff (Allan et al., 2017), and exploitative labor





conditions (El Chami et al., 2020). In addition, sugarcane receives a disproportionate amount of policy support in many coun-
tries through mechanisms such as market price support, ethanol mandates, and assistance to sugar mills. According to recent
OECD estimates, sugar receives commodity-specific transfers of more than 20% of farm receipts globally, higher than any
other food commodity (OECD, 2023). In some countries, this share is much higher, such as Mexico (37%), the United States
(48%), Indonesia (55%), and the Philippines (62%) (OECD, 2023).
Despite the prominent role of sugarcane in many economies and the key support from government, few countries provide
timely information on the status and dynamics of sugar cultivation. Such information could be helpful in studying the full
effects of sugar cultivation on the health of both humans and the environment, thus informing public policy. Better data could
also help aid sugar producers in their attempts to optimize productivity and profits, for example by helping to better understand
factors that determine yield variation.
In an effort to fill the significant data gaps relating to sugarcane cultivation, we present here an approach and dataset that
uses satellite remote sensing to map precise locations of sugarcane canopies around the world. Remote sensing has long been
used to map areas of individual crops, with several countries producing annual, publicly available maps of crop types based
on satellite data, such as the Cropland Data Layer (CDL) in the United States (Boryan et al., 2011) and the Annual Crop
Inventory in Canada (Agriculture and Agri-Food Canada). Yet these maps have historically required ground data to calibrate
the satellite models each year, which precludes their use in countries without a concerted government effort to maintain ground
data collection.
Rather than rely on ground data, our approach relies on two features of sugarcane that together make it a unique crop
throughout most of the regions where it is grown – it is much taller than most crops (often exceeding 3 meters in height), and
grows across multiple years. In recent work (Di Tommaso et al., 2021, 2023), we demonstrated the ability of lidar measurements
acquired by the Global Ecosystem Dynamics Investigation (GEDI) (Dubayah et al., 2020) to identify tall canopies within
agricultural landscapes. Here we extend that work to map tall crops in each month over a four year period, and then identify
sugarcane fields as those that are tall for a sufficiently large fraction of the study period. We find that this approach is able to
map sugarcane with impressive detail across a wide number of countries, using both government statistics and independent
maps in some countries to evaluate our product.

## 2   Datasets

The datasets utilized in this study include:
1. GEDI and Sentinel-2 Sensors: Data from the Global Ecosystem Dynamics Investigation (GEDI) and Sentinel-2 (S2)
satellite sensors were employed for data acquisition. Pre-processing steps were taken to prepare these datasets for analysis.
2. Land Cover Products: Various land cover products were employed to delineate the cropped areas within the study area.
3. Calibration and Validation Datasets: Specific datasets were utilized for the calibration and validation of the sugarcane
maps generated in this study.



## 2.1 GEDI data

GEDI, a sensor mounted on the International Space Station (ISS), captures lidar waveforms within the latitudinal range of 51.6°
N to 51.6° S to analyze the Earth's surface in three dimensions. It is the first spaceborne lidar instrument specifically designed
for assessing vegetation structure (Dubayah et al., 2020). Equipped with three lasers emitting near-infrared light at 1064 nm
wavelength, GEDI features two full-power lasers along with a third laser divided into dual beams, generating a total of four
beams. Through optical dithering across-track, each beam creates eight ground tracks (comprising four full-power tracks and
four cover tracks) spaced 600 meters apart on the ground. The shots produced have an average footprint diameter of 25 meters
and are separated by 60 meters along-track.

For this study, we used the GEDI dataset Level 2A (L2A) and Level 2B (L2B) from April 2019 to December 2022, available
in GEE data catalog.

The Level 2 data offer insights into the vertical canopy distribution derived from waveform returns at the footprint level. Our
primary dataset was GEDI's L2A Geolocated Elevation and Height Metrics Product, primarily comprising Relative Height
(RH) metrics. These RH metrics collectively characterize the waveform data acquired by GEDI, providing information about
the height at which a specific percentage of energy is returned relative to the ground. RH values are reported at 1% intervals,
resulting in a total of 101 metrics. Additionally, we used the L2B dataset to extract the GEDI view angle at each shot location,
specifically using the 'local beam elevation' property. This information was used to filter out GEDI shots with a view angle
below 1.51 rad, to avoid classification errors, as recommended in Di Tommaso et al. (2023).

The GEDI L2A dataset (`LARSE/GEDI/GEDI02_A_002_MONTHLY`) and L2B dataset (`LARSE/GEDI/GEDI02_B_002_MONTHLY`)
represent a rasterized version of the original GEDI products, where each GEDI shot footprint is depicted by a 25-meter pixel
(Healey et al., 2020). This rasterization process, however, may introduce an additional geolocation error beyond the initial
GEDI shot error. The raster images are structured as monthly composites of individual orbits conducted during the respective
month (refer to Figure 1). Within these raster images, RH values, along with quality flags and metadata, are preserved as raster
bands.

## 2.2 Sentinel-2

We employed the S2 surface reflectance Harmonized collection, which is readily available in the Google Earth Engine (GEE)
platform. Clouds were filtered out using the S2 Cloud Probability dataset provided by SentinelHub in GEE. Utilizing this
dataset, we generated yearly (January to December) time series for each pixel. These time series were then utilized to compute
harmonic features, with an order of n=3 and omega=1, for a combination of bands including 'NIR' (Near Infrared), 'SWIR1'
(Shortwave Infrared 1), 'SWIR2' (Shortwave Infrared 2), 'RDED4' (Red Edge Band 4), and 'GCVI' (Green Chlorophyll
Vegetation Index) (Gitelson et al., 2005). This approach, proven successful in previous studies, has demonstrated efficacy in
tasks related to crop type classification. GCVI is computed as

$GCVI = NIR/Green - 1$

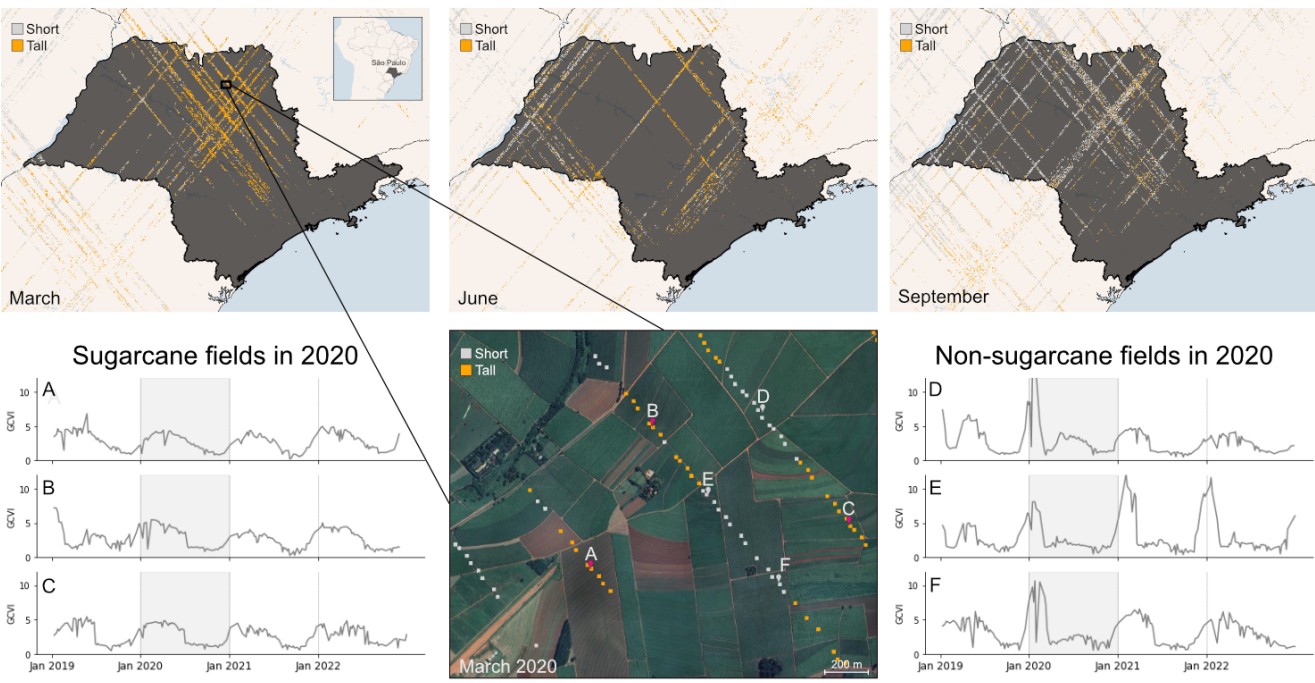

**Figure 1.** GEDI shots over São Paulo, the main sugarcane-producing state in Brazil. Top panel shows GEDI coverage in three different months, March, June and September, over the 4 years of data. Shots represented by a 25 m pixel are color-coded according to the short/tall classification by the GEDI model. Gaps in GEDI shot orbits may be attributed to quality issues. For instance, in June 2020, a significant portion of shots experienced low view angles and were subsequently filtered out, resulting in sparser GEDI coverage during this period, as illustrated in the top-middle panel. Additionally, in this region, there appears to be a higher proportion of shots classified as tall around the beginning of the year compared to later months. Bottom-middle panel show a zoom in at field level (© Google Earth Engine). GCVI S2 time series from 2019 to 2022 over sugarcane (on the left) and non-sugarcane fields (on the right), with the year 2020 highlighted in gray shading. GEDI accurately identifies tall fields that are growing sugarcane in March (A,B,C), and short fields that are not growing sugarcane in 2020 (D,E,F).





For each spectral band or vegetation index $f(t)$, the harmonic regression takes the form
$$f(t) = c + \sum_{k=1}^{n} [a_k \cos(2\pi\omega kt) + b_k \sin(2\pi\omega kt)]$$

where $a_k$ are cosine coefficients, $b_k$ are sine coefficients, and $c$ is the intercept term. The independent variable $t$ represents
the time an image is taken within a year expressed as a fraction between 0 and 1. The number of harmonic terms $n$ and the
periodicity of the harmonic basis controlled by $\omega$ are hyperparameters of the regression. This resulted in seven features per
band, for a total of 35 coefficients. These estimated values represent the S2-based harmonic features used in the subsequent
classification process.

## 2.3 Crop mask

Despite the abundance of global and regional cropland maps, considerable uncertainties and discrepancies persist regarding
both the total area and spatial distribution. To identify cropped areas comprehensively, we conducted an analysis encompassing
all global land cover products detailed in Kerner et al. (2023). Through visual inspection and subsequent examination of the
datasets outlined later, we observed that relying solely on a single product often resulted in the underestimation of cropland
area in certain regions, while another product exhibited similar limitations elsewhere. Recognizing the inherent risk of inac-
curate crop masks leading to either over- or underestimation, we opted to ensure a more robust global coverage by integrating
information from three distinct global land cover products. We defined a pixel as cropland if any of the three maps classified it
as such. This approach, involving the combination of these datasets, enabled us to enhance the completeness of cropland areas
worldwide.
The three global products are: the European Space Agency (ESA) WorldCover 2020 (Zanaga et al., 2021), ESRI 2020 global
Land Use Land Cover (Karra et al., 2021) and the 2019 GLAD Global Cropland Maps (Potapov et al., 2022).
A visual example of the three crop masks is provided for Brazil in Figure 2.
The ESA and ESRI 2020 products provide a global land cover map for 2020 at 10 m resolution, the former based on
Sentinel-1 and Sentinel-2 data, and the latter based on Sentinel-2 alone. Maps are available in the Google Earth Engine (GEE)
(Gorelick et al., 2017) official and community data catalogs, respectively (Roy et al., 2024). The 2019 GLAD Map provides
binary cropland classifications at 30 m. Classification is performed using bagged decision trees with features extracted from
time series of Landsat Analysis Ready Data (ARD).
Divergences exist among these land cover and land use products regarding the categorization of croplands, particularly
concerning the inclusion of tree crops. ESA WorldCover encountered issues such as underestimation of cropland areas in Brazil
and Africa, particularly in fragmented regions with mixed land covers. Contrarily, the WorldCover 2020 product identified more
tree cover, representing orchards, compared to other ESRI products.
ESA's definition of cropland encompasses land that is covered with annual crops sowed and harvested at least once within
12 months after the sowing date. This cropland typically produces an herbaceous cover and may include some tree or woody
vegetation but excludes perennial woody crops. ESRI defines croplands as human-planted cereals, grasses, and crops not at tree





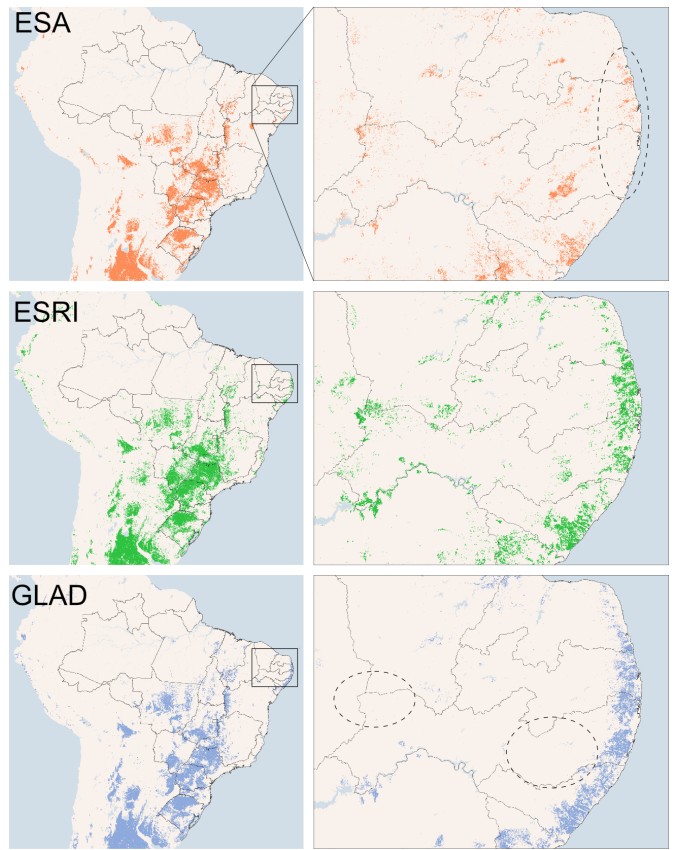

**Figure 2.** An example of difference in crop masks in Brazil (© Google Earth Engine). Dotted circles highlight areas of disagreement between maps. The ESA crop mask exhibits omissions in cropland detection in North-Eastern Brazil, whereas ESRI and GLAD capture more cropland in this region. ESRI tends to overmap cropland, often including orchards, while GLAD exhibits a more conservative approach, albeit missing some cropland in areas where both ESA and ESRI map it.

height, including rice paddies and irrigated agriculture, while GLAD excludes perennial woody crops and permanent pastures from its definition, focusing on herbaceous crops for human consumption, forage, and biofuel.

## 2.4 Calibration and Validation datasets

Below we describe the datasets used for calibrating and validating the sugarcane maps. The data pertain to the main sugarcane-producing countries according to the Food and Agriculture Organization (FAO) (Food and Agriculture Organization's Statistical Database (FAOSTAT)), as presented in Table 1. Within each section, the countries are in decreasing order of sugarcane production.



**Table 1.** Main sugarcane-producing countries according to FAO 2022.

| Rank | Area | Production (million tonnes) | Production (%) | Area harvested (million hectares) | Area harvested (%) |
|------|------|-----------------------------|----------------|-----------------------------------|--------------------|
| 1 | Brazil | 724 | 37.7 | 9.9 | 37.8 |
| 2 | India | 439 | 22.9 | 5.2 | 19.8 |
| 3 | China | 103 | 5.4 | 1.3 | 5.0 |
| 4 | Thailand | 92 | 4.8 | 1.5 | 5.8 |
| 5 | Pakistan | 87 | 4.6 | 1.3 | 5.1 |
| 6 | Mexico | 55 | 2.9 | 0.8 | 3.1 |
| 7 | Colombia | 35 | 1.8 | 0.4 | 1.4 |
| 8 | Indonesia | 32 | 1.7 | 0.5 | 1.9 |
| 9 | USA | 31 | 1.6 | 0.4 | 1.4 |
| 10 | Australia | 28 | 1.5 | 0.3 | 1.3 |
| 11 | Guatemala | 26 | 1.4 | 0.2 | 0.9 |
| 12 | Philippines | 23 | 1.2 | 0.4 | 1.5 |
| 13 | South Africa | 17 | 0.9 | 0.3 | 1.0 |
|  | Rest of the world | 224 | 11.7 | 3.6 | 13.9 |

The list includes field-level labels, raster datasets, and government-reported sugarcane area data at administrative level 2 or
3. The specific sources for these datasets may vary depending on the region and the year of data collection. A summary of all
data available by region is provided in Table 2.
**2.4.1   Point level data**
The WorldCereal "sv_croptype_validations" dataset (Lesiv et al., 2023) includes observations of crop types in 2021 and 2022
at global scale along with their coordinates. This dataset was compiled and released by WorldCereal through a meticu-
lous process involving expert manual labeling. Utilizing an IIASA tool known as "Street Imagery validation" (accessible at
https://svweb.cloud.geo-wiki.org/), contributors were able to examine street-level images, including those from platforms like
Google Street View and Mapillary, and accurately identify crop types. It's important to note that this dataset is entirely distinct
and separate from existing maps and reference datasets, providing an independent source of valuable information for agri-
cultural analysis. The dataset contains labels for various crop types, including sugarcane, for several countries of interest. In
Brazil, sugarcane is the most prevalent crop label, accounting for 1.6k labels, followed by maize (~910) and soybean (~550).
In Mexico crop labels alongside sugarcane ( 50 labels) include maize (~40). Australia's crop distribution includes wheat (~120
labels) and sugarcane (~20). Meanwhile, in the Philippines, rice (~80 labels) is prevalent alongside sugarcane (~70).
Other countries represented in the WorldCereal dataset with a smaller number of samples include China, Colombia, India,
Pakistan, South Africa and Thailand.



**Table 2.** Summary of all available dataset by country and data type. The datasets used for calibrating our method are marked with an asterisk. WorldCereal point data refer to the years 2021-2022.

| Country | Raster | Field points | Government statistics |
|---|---|---|---|
| Brazil | (binary) 2018-2019 | World Cereal* | 2022 |
| India | | Plantix 2020-2021* <br> World Cereal | 2019-2020 |
| China | (binary) 2019-2020* | World Cereal | 2022 |
| Thailand | | GSV points 2022* <br> World Cereal | 2022 |
| Pakistan | | World Cereal | 2021-2022 |
| Mexico | | World Cereal* | 2022 |
| Colombia | | World Cereal | 2019 |
| Indonesia | | | 2021 |
| USA | CDL 2019-2022* | | 2018 |
| Australia | | World Cereal* | 2020-2021 |
| Guatemala | | | 2003 |
| Philippines | | World Cereal* | 2021 |
| South Africa | SANLC 2020* | SANLC points 2020 <br> World Cereal | 2017 |

In India we accessed crop type labels crowdsourced from farmers via Plantix, a free Android application developed by
Progressive Environmental and Agricultural Technologies (PEAT). The Plantix app is used by farmers who upload photos of
their crops to seek assistance in diagnosing and treating crop diseases. As part of the disease diagnosis process, PEAT uses
a convolutional neural network to assign crop labels based on the submitted photos. We used labels for the years 2020 and
2021 in the Indian states of Maharashtra and Uttar Pradesh (UP), where the accuracy of Plantix crop type labels exceeds 90%
for most major crops. Data have been cleaned to remove location inaccuracy (keeping only submissions with GPS accuracy
better than 10 m), as suggested by previous work by Wang et al. (2020). Additionally, to mitigate any bias, Plantix labels were
sampled to match the proportion of government-reported crop areas by crop, as certain labels, such as those for vegetables,
were more prevalent due to their susceptibility to diseases.
In Thailand we accessed crop type labels obtained with Google Street View (GSV) (Laguarta et al., 2023) for the year 2022.
These labels were generated by combining deep learning and street view imagery over Thailand, requiring minimal manual
labeling. Labels include sugarcane, cassava, maize, rice, and an "other" crop class. Labels accuracy XX To ensure the labels
were representative of the landscape, they were sampled in alignment with government-reported crop areas.
In South Africa, independent reference points, used for validating the South African National land cover 2020 (SANLC
2020) map, are provided by the Department of Forestry, Fisheries and the Environment (South Africa - DFFE).



### 2.4.2 Raster data

In Brazil and China, sugarcane masks at 30m resolution were recently published by Zheng et al. (2022a) and Zheng et al. (2022b). These maps were generated using a time-weighted dynamic time warping method. In Brazil, maps are available for 14 states for 2016–2019, with a reported overall accuracy for the year 2018 of 91%, and user's and producer's accuracies reaching 94% and 87%, respectively.

In China, maps are available for 2016–2020 for four southern provinces, which map over over 95% of the sugarcane cultivation areas in China: Guangxi (64%), Yunnan (18%), Guangdong (12%), and Hainan (1%) provinces. The reported overall accuracy for the year 2019 is 92.7%, with reported user's and producer's accuracies of 85.6% and 86.7%.

The Cropland Data Layer (CDL) (Boryan et al., 2011) produced by the United States Department of Agriculture (USDA) provides yearly crop type maps across the conterminous US at 30 m spatial resolution. Maps are based on Landsat and other satellite imagery using training data from the Farm Service Agency (FSA). Sugarcane plantations in the contiguous United States are primarily concentrated in three states: Florida, Louisiana, and Texas. Accuracy of CDL on FSA labels are available in the CDL metadata, with precision and recall for sugarcane in 2019-2022 exceeding 72%, 94% and 93% in Texas, Florida, and Louisiana, respectively.

The South African National Land Cover 2020 (SANLC 2020), recently published by the Department of Forestry, Fisheries, and the Environment (South Africa - DFFE), was generated at a 20-meter resolution utilizing S2 imagery. The overall accuracy of this land cover classification is 85.5%. The accuracy for the sugarcane classes surpasses 95% for user's accuracy and 82% for producer's accuracy.

### 2.4.3 Government statistics

The Brazilian Institute of Geography and Statistics (IBGE) (Instituto Brasileiro de Geografia e Estatística) offers comprehensive data on various agricultural metrics, including the planted and harvested areas, production volumes, and average yields, on an annual basis for agricultural commodities. In our research, we utilized the municipality-level (admin 2) data for sugarcane planted and harvested areas for the latest available year, 2022.

In India, the Ministry Of Agriculture and Farmers Welfare releases crop production statistics (Indian Department of Agriculture) at the district level (admin 2). For our analysis, we incorporated district-level crop area statistics for the most recent available year, which is the 2019–2020 growing season.

In China, the Statistical Yearbooks serve as annual publications providing comprehensive insights into the economic and social development of each province. These publications encompass data from the previous year, offering statistics at both the provincial level and the local levels of cities (level 2). For our analysis, we obtained sugarcane sown area data from the Statistical Yearbook for the 2022 growing season for the four sugarcane producing provinces: Guangdong (Guangdong Provincial Bureau of Statistics), Guangxi (Statistics Bureau of Guangxi Zhuang Autonomous Region), Yunnan (Yunnan Provincial Bureau of Statistics), and Hainan (Hainan Provincial Bureau of Statistics).





The agricultural statistics of Thailand for the year 2022, including data on sugarcane harvested area, were sourced from the
relevant government authority at province level (admin 1) (Office of Agricultural Economics).
The district-wise statistics on crops area and production for the growing season 2021-22 in Pakistan were obtained from
the government of Pakistan at district level (admin 3) (Ministry of National Food Security and Research). Due to uncertainties
regarding district borders over time, the data were processed and aggregated at level 2 to ensure consistency and accuracy in
the analysis.
The annual agricultural statistics provided by the Government of Mexico (Agri-Food And Fisheries Information Service)
encompass a wide range of information, including data on planted area, harvested area, damaged area, average rural prices,
volume, and value of production for both cyclical and perennial crops, categorized by water modality. These reports cover all
32 federal entities of the country, with detailed breakdowns at the national, state, district, and municipal levels (admin 2). For
our analysis, we specifically extracted sugarcane area data at the municipality level for the year 2022.
The National Agricultural and Livestock Survey Survey (ENA) conducted in 2019 (National Administrative Statistics De-
partment), provides data on sugarcane planted area, production, and yield by region (admin 1) for the year 2019 in Colombia.
The Indonesia Central Statistics Agency (Badan Pusat Statistik (BPS)) serves as the official statistical agency of the Indone-
sian government, tasked with collecting, processing, analyzing, and disseminating statistical data and information throughout
the nation. It provides comprehensive statistics on plantation area by province (admin 1), which offers insights into the distri-
bution of agricultural land across different regions of Indonesia. Specifically, the dataset comprises the area of annual crops
such as oil Palm, coconut, rubber, coffee, cocoa, and tea, representing the planted area at the end of the year. Additionally,
the dataset includes information on seasonal crops like tobacco and sugarcane, with data reported as the monthly cumulative
harvested area. The most recent report refers to the year 2021.
In the United States, county-level (admin 2) statistics on sugarcane area are available from the United States Department of
Agriculture's National Agricultural Statistics Service (NASS) (USDA Natlional Agricultural Statistics Service). We accessed
the most recent data for counties in the key sugarcane-producing states Florida, Louisiana, and Texas for the year 2018 using
the NASS Quickstats database.
Statistics on the production of agricultural commodities, encompassing cereal and broadacre crops, fruit and vegetables, and
livestock on Australian farms, are provided by the Australian government (Australian Bureau of Statistics). These statistics are
made available on a yearly basis, with the most recent data available for the 2020-2021 growing season (at Statistical Areas
Level 2).
In Guatemala, data on sugarcane production by department (admin 1) for the agricultural year 2002/2003 was obtained from
the IV National Agricultural Census (Guatemala Nationl Institue of Statistics). Sugarcane accounted for 28.4% of the total area
cultivated with permanent and semi-permanent crops. The department of Escuintla recorded the highest sugarcane productions
for the census year, comprising 87.7% of the total production.
The Philippine Statistics Authority (PSA) (Philippine Statistics Authority) releases annual provincial statistics on Agriculture
and Fisheries. These statistics include the total area of sugarcane and the percent distribution of sugarcane production by
region. Although direct access to sugarcane area by region is not available, an approximation can be made by assuming that





the percentage of production falls within the same range as the percentage of area by region. The most recent year for which
data is available is 2021.
The Statistics department of South Africa (Statistics Department - South Africa) conducts the Census of Commercial Agri-
culture, 2017 (CoCA 2017), which publishes results at the municipal level (admin 3). The primary objective of this survey is to
gather financial, production, employment, and related information pertaining to the commercial agriculture industry in South
Africa. It is important to note that CoCA 2017 only covers enterprises registered for value-added tax (VAT). Consequently, the
census does not include smallholder farming. Instead, it utilizes VAT records as a sampling frame, thereby excluding entities
that are non-VAT registered. It is noteworthy that commercial farmers account for 80% of the country's agricultural value.

## 3 Methods

### 3.1 Sugarcane phenology

Sugarcane is primarily grown in tropical and sub-tropical regions of the world. It is a tall semi-perennial crop, with a growth
cycle lasting typically between 12 to 18 months before it is ready for harvesting. The specific duration of this cycle varies
depending on factors like the sugarcane variety, local climate, and geographical conditions in each region. After the first
harvest, sugarcane can regrow from the same root systems for multiple years (ratoon crops), resulting in subsequent yield
losses due to a reduction in stalk population. To ensure sustainable yields and maintain soil fertility, sugarcane areas are often
rotated with other crops to aid in nitrogen fixation for subsequent sugarcane growth seasons. Cultivation practices also involve
planting different sugarcane varieties within the same plantations to minimize susceptibility to diseases. Figure 1 provides a
visual example of sugarcane time series in Brazil and rotation with soybean.

### 3.2 Area of interest

We initiated our study by focusing on the main sugarcane-producing countries listed in Table 1.
We established a $2° \times 2°$ grid overlaying these countries. To reduce computation, grid cells were selected based on two
criteria: a cropland coverage exceeding 1%, determined using the European Space Agency (ESA) Crop Mask dataset, and
a sugarcane area greater than 0, derived from the Spatial Production Allocation Model (SPAM) (International Food Policy
Research Institute, 2019). These selected grid cells represent the regions where we aimed to predict sugarcane presence.

### 3.3 GEDI data processing

All GEDI shots from April 2019 to December 2022 over cropland pixels, passing over these $2° \times 2°$ grid cells, were classified
as either short, tall, or tree by a GEDI model trained in Di Tommaso et al. (2023). Each classification was accompanied by a
confidence value.
Prior to further analysis, the predicted shots underwent a filtering process to retain only high-quality data. Initially, shots
were filtered based on the quality and degrade flags provided as properties in the GEDI dataset. Additionally, predictions with





confidence scores lower than 0.8 were discarded. A crucial step involved filtering out shots with low view angles and those
over high-slope terrain, defined as areas with slopes exceeding $5°$, as both factors can impact the accuracy of GEDI model
predictions. View angle information was retrieved from the L2B dataset, enabling the exclusion of low view angle shots.
Furthermore, we opted to exclude shots identified as belonging to the tree class by the GEDI model. This decision was
motivated by the likelihood that such shots may encompass a mixture of crops and trees within the GEDI footprint, which, at
a diameter of 25 meters, surpasses the size of the 10-meter S2 pixel by over four times.
Figure 1 shows the spatial coverage of GEDI over time and the changing proportion of tall and short labels over cropland.

### 3.4 S2 model training and classification

Utilizing the GEDI predictions as binary labels, we trained separate local S2 models for each grid cell and for each month
of the year. We opted for a random forest model for its well-documented advantages, including high accuracy, computational
efficiency, and smooth integration into large-scale applications within GEE. The S2 models were trained using S2 harmonics
coefficients as features and the GEDI predictions as labels. For each grid cell, we aggregated GEDI labels for each month across
different years and extracted the corresponding S2 features for the same year as the GEDI label. Subsequently, we constructed
pooled models for each month and generated predictions for four years, utilizing features specific to each year. This process
yielded 48 monthly predictions for each grid cell, where each 10 m by 10 m pixel within the crop mask was classified as either
short or tall.
In order to reduce spatial artifacts during the mosaicking of adjacent cells, we created predictions for pixels in a $0.2°$ buffer
around each cell. Subsequently, on a monthly basis, we mosaicked the overlapping predictions, selecting the predictions from
the cell with the higher GEDI-S2 kappa score.

### 3.5 Calibration/Sugarcane identification

We determined sugarcane presence by computing the frequency of tall predictions for each pixel across the 48 monthly predic-
tions. Pixels were classified as sugarcane if the frequency of tall predictions exceeded a certain threshold.
To distinguish sugarcane from other tall crops, a single threshold across all countries is not optimal, since the threshold
will depend on the mix of crops alongside sugarcane, the phenological characteristics of both sugarcane and other crops, and
agricultural management practices.
The selection of the threshold was guided by a calibration approach based on available *in situ* data. To determine the threshold
in countries where we had large number of labels of sugarcane and different crop type classes, we used point level calibration
and relied on the threshold that produced the highest kappa score.
$$Kappa\ score = \frac{P_o - P_e}{1 - P_e}$$
where
– $P_o$ is the proportion of observed agreement, i.e. the accuracy achieved by the model

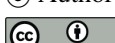



– $P_e$ is the proportion of agreements expected by chance
This methodology was applied in Brazil, India, Thailand, and South Africa, where we had many (> 600) ground samples.,
as well as in China and the USA where point labels were unavailable but crop type maps developed through a combination
of ground and satellite data were accessible. In these cases, samples were obtained by random sampling of the reference crop
maps.
In other countries with a limited availability of sugarcane labels (n<200), we extracted at each location of a sugarcane
label the number of tall months in our map, and then calculated the 10th percentile of this value across all such locations.
This threshold therefore ensures that 90% of the reference sugarcane labels would be classified as sugarcane. In countries
where ground labels were lacking, we set the threshold equal to that of a nearby country, based on the assumption that the
characteristics of the sugarcane were most similar in nearby locations.

### 297  3.6  Validation

To validate our sugarcane maps, we compared them against a combination of available point samples, raster maps of crop
type, and reported sugarcane area of government statistics. Because of the nature of sugarcane, a semi-perennial crop, we are
mapping total-stable sugarcane area in the 4-year period. Although we do not expect perfect agreement against government
reported planted or harvested area for a single year, a comparison with government data still provides a useful assessment of
how well our maps capture broad spatial patterns.

### 303  4  Results

We first present the outcomes of the calibration strategy, outlining the optimal threshold for sugarcane identification based
on available data specific to each country. For validation purposes, we compare the results against field points and rasters
and assess the sugarcane area against government-reported data. These evaluations are conducted for each country using the
selected threshold and employing a combined ESA and GLAD crop mask. We find that combining these two maps helps cover
the majority of cropland in most regions while avoiding the mapping of orchards that are often included in the ESRI crop mask.
It's worth noting that even though results are provided for the cropland area mapped in the ESA and GLAD masks, a sugarcane
map is produced for the area covered by the ESRI crop mask as well, and it is made available in our dataset. Further details
about the data release are provided in the data availability section.

### 312  4.1  Calibration

For calibration, we employed various strategies due to the absence of in situ labels across all countries of interest. Results of
the calibration for countries with abundant ground samples are illustrated in Figure 3.
China, Thailand, and South Africa exhibit low sensitivity to the chosen threshold. In Thailand, the threshold is optimized to
avoid mostly confusion with cassava, a shrubby perennial that is usually 2–3 m in height. In South Africa, most of the point



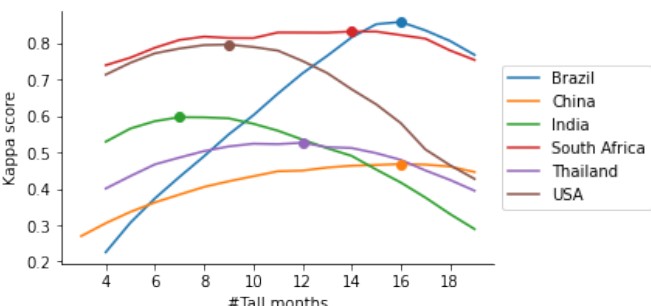

**Figure 3.** Identification of the threshold for classifying pixels as sugarcane based on the kappa score for countries with abundant in situ crop type labels. X-axis shows number of months (out of a total of 48) where a tall crop was present, and y-axis shows the kappa score for a model that classifies as sugarcane all pixels with at least this many months with tall crops. Dots represent the maximum kappa score value and the chosen threshold for each country. In India, 7 and 8 month thresholds were tied and we opted to use a threshold of 8 to be conservative and prioritizing higher precision in our map, avoiding the inclusion of other crops.

samples are sugarcane, followed by small-scale and commercial annual crops. The optimal threshold helps to avoid confusion
with commercial irrigated annual crops.
The threshold selection is critical in Brazil, mostly to avoid confusion with maize, another tall crop ranging from 1.2–4 m in
height.
India and the US exhibit moderate sensitivity and lower optimal thresholds, perhaps due to shorter sugarcane phenological
cycles and the absence of other crops that appear tall in sugarcane growing areas. In India, the calibration curve appears very
flat between 7 and 8 months. To err on the side of caution, and avoid including other crops in our sugarcane map, we chose
to set the threshold at 8 months. In the US, the threshold serves to avoid confusion with maize, which is present but not as
common as in Brazil.
In regions where insufficient labels were available for crop types other than sugarcane to compute a reliable kappa score, such
as Mexico, Australia, and the Philippines, we adopted the 10th percentile approach. Conversely, in regions where no data were
accessible, we determined the threshold based on the neighboring country. This last strategy was applied in Pakistan, Colombia,
Indonesia, and Guatemala. Results of the chosen calibration method and threshold for all the countries are summarized in Table
3.

### 4.2    Sugarcane Maps

Sugarcane maps for the main producing countries obtained applying the calibration threshold previously identified, across the
48 monthly predictions are shown in Figure 4.





**Table 3.** Summary of the thresholds used for calibrating the sugarcane maps. The threshold is expressed as the number of months over a 48 month-period. Diverse metrics and data sources have been adopted across different countries as a result of disparities of *in situ* data availability. Rows marked with N/A denote the absence of available data, and a threshold from a neighboring country was adopted. Specifically, Pakistan employed the same threshold as India, Colombia as Mexico, Indonesia as Thailand and Guatemala as Mexico. Threshold range from as low as 8 months in India, Pakistan and the Philippines, to as high as 16 months in Brazil and China. These disparities reflect differences in sugarcane phenology, management practices, and co-cultivation with other crops, tall or short.

| Rank | Country | Data Source | Metric | Threshold |
|---|---|---|---|---|
| 1 | Brazil | WorldCereal | kappa | 16 |
| 2 | India | Plantix points | kappa | 8 |
| 3 | China | Raster | kappa | 16 |
| 4 | Thailand | GSV points | kappa | 12 |
| 5 | Pakistan | N/A | | 8 |
| 6 | Mexico | WorldCereal | 10th perc | 14 |
| 7 | Colombia | N/A | | 14 |
| 8 | Indonesia | N/A | | 12 |
| 9 | USA | CDL | kappa | 9 |
| 10 | Australia | WorldCereal | 10th perc | 11 |
| 11 | Guatemala | N/A | | 14 |
| 12 | Philippines | WorldCereal | 10th perc | 8 |
| 13 | South Africa | SANLC points | kappa | 14 |

## 4.3 Validation

### 4.3.1 Validation against field points

We provide a summary of point-level validation results for the sugarcane maps by country based on field-level data in Figure 5.

Performance metrics vary across countries, with F1 scores for sugarcane exceeding 0.8 for most countries. Notably, Brazil, Mexico, Australia, the Philippines, and South Africa exhibit strong performance, with F1 scores higher than 0.9. However, exceptions are observed in certain regions.

In Thailand, utilizing GSV samples yields an F1 score on sugarcane of 0.57, with precision and recall scores of 0.53 and 0.62, respectively. The predominant confusion is observed with the cassava class. This is not surprising given their coexistence in similar geographic regions and that cassava plants can grow over 2 m. It is also common for farmers to alternate between cassava and sugarcane cultivation in their fields. In contrast, performance in Thailand using WorldCereal data appears to be better, but it is essential to note that cassava is not included in this dataset. WorldCereal crop classes in Thailand include rice, sugarcane, and maize. Additionally the number of WorldCereal samples (75) is substantially limited compared to GSV samples (~19k).



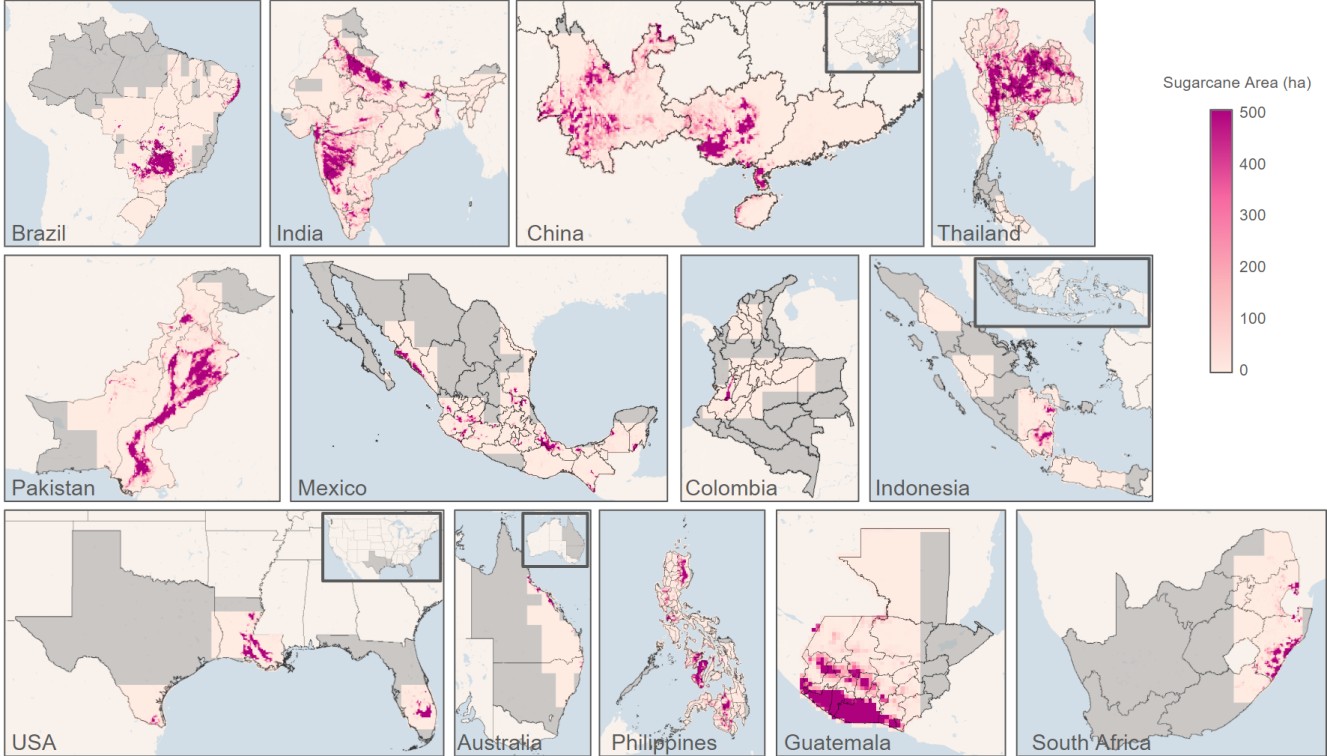

**Figure 4.** Sugarcane maps for top 13 producing countries (© Google Earth Engine). For visualization, the original 10 m maps were resampled at 10 km resolution and show the sugarcane area in hectares for each 10 × 10 km pixel (1000 ha). The area that we did not process–due to lack of cropland or sugarcane–is colored in gray. For China, Indonesia, the US, and Australia, the highlighted gray area in the inset indicates the regions for which zoom-ins are provided.

In India, contrasting results are observed between different datasets. For instance, using Plantix labels yields an F1 score of
0.67 for sugarcane, with precision and recall at 0.65 and 0.69, respectively. Notably, performance in Maharashtra (MH) lags
behind Uttar Pradesh (UP), with F1 scores of 0.56 and 0.7, respectively. The lower performance of MH is mostly due to low
precision (0.5), caused from misclassification of maize as sugarcane. Conversely, utilizing WorldCereal data in India results
in an F1 score of 0.82 for sugarcane, with precision and recall metrics of 1 and 0.69, respectively. This is explained by fewer
maize labels, with labels for the other class including mostly rice and wheat. It's worth noting in this case as well the limited
number of WordCeral samples (115) in this region compared to Plantix (~37k).
Similarly, Pakistan, using WorldCereal labels, exhibits an F1 score of 0.6, primarily attributed to low recall (0.43).
**4.3.2  Validation against raster datasets**
We offer a visual comparison between reference maps and predicted sugarcane maps for regions where crop type maps are
available, depicted in Figure 6. In cases where multiple years of sugarcane maps were accessible but did not correspond to the

Earth System
Science
Data

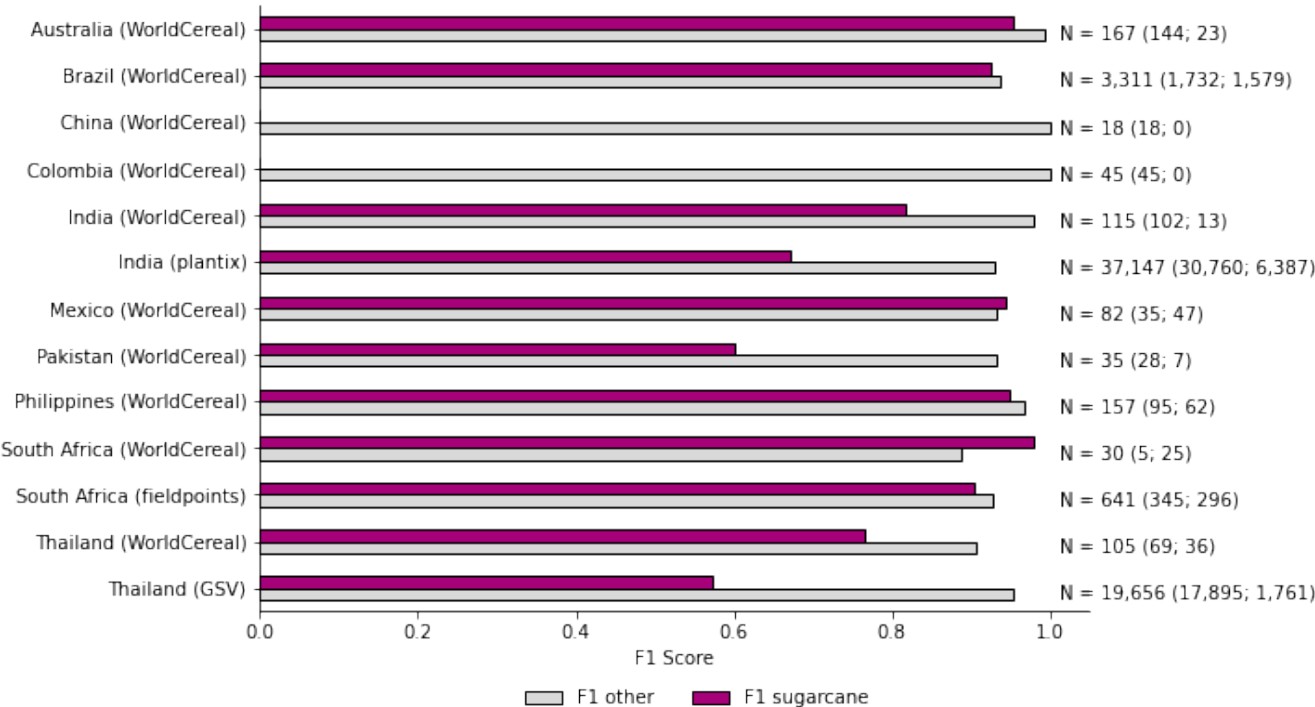

**Figure 5.** Results of point level validation. In parenthesis are reported the numbers of samples labeled non-sugarcane and sugarcane labels.

same years as our study, as in the case of Brazil and China, we utilized the two most recent years. Sugarcane was classified
as present in a pixel if it appeared as sugarcane at any point during these years, accounting for potential crop rotation. For the
US, where the Cropland Data Layer (CDL) is available annually from 2019 to 2022, we considered a pixel as sugarcane if it is
classified as sugarcane for at least two years out of the four.

To evaluate a measure of agreement between maps, we randomly sampled 10k cropland points for each state/admin1 covered
by the raster maps and reported F1 scores in Fig. 6. These metrics pertain to the entire mapped raster area, not just the portion
depicted in the zoomed-in view in the figure. Across different regions, F1 scores for sugarcane varied, ranging from 0.47 in
China to 0.84 in the USA.

In Brazil, the raster encompasses 13 states, with a relatively lower F1 score of 0.6 for sugarcane. This discrepancy is reflected
in the precision of 0.55 and recall of 0.66. However, in São Paulo, the F1 score improves to 0.74, characterized by higher
precision (0.82) and recall (0.67).

In China, the overall F1 score of 0.47 is derived from data spanning all four provinces. Notably, in Guangxi, the primary
sugarcane-producing region, the F1 score increases to 0.64, with the same precision and recall (both 0.64).

In the USA, precision and recall values stand at 0.85 and 0.82, respectively, indicating strong agreement between maps.

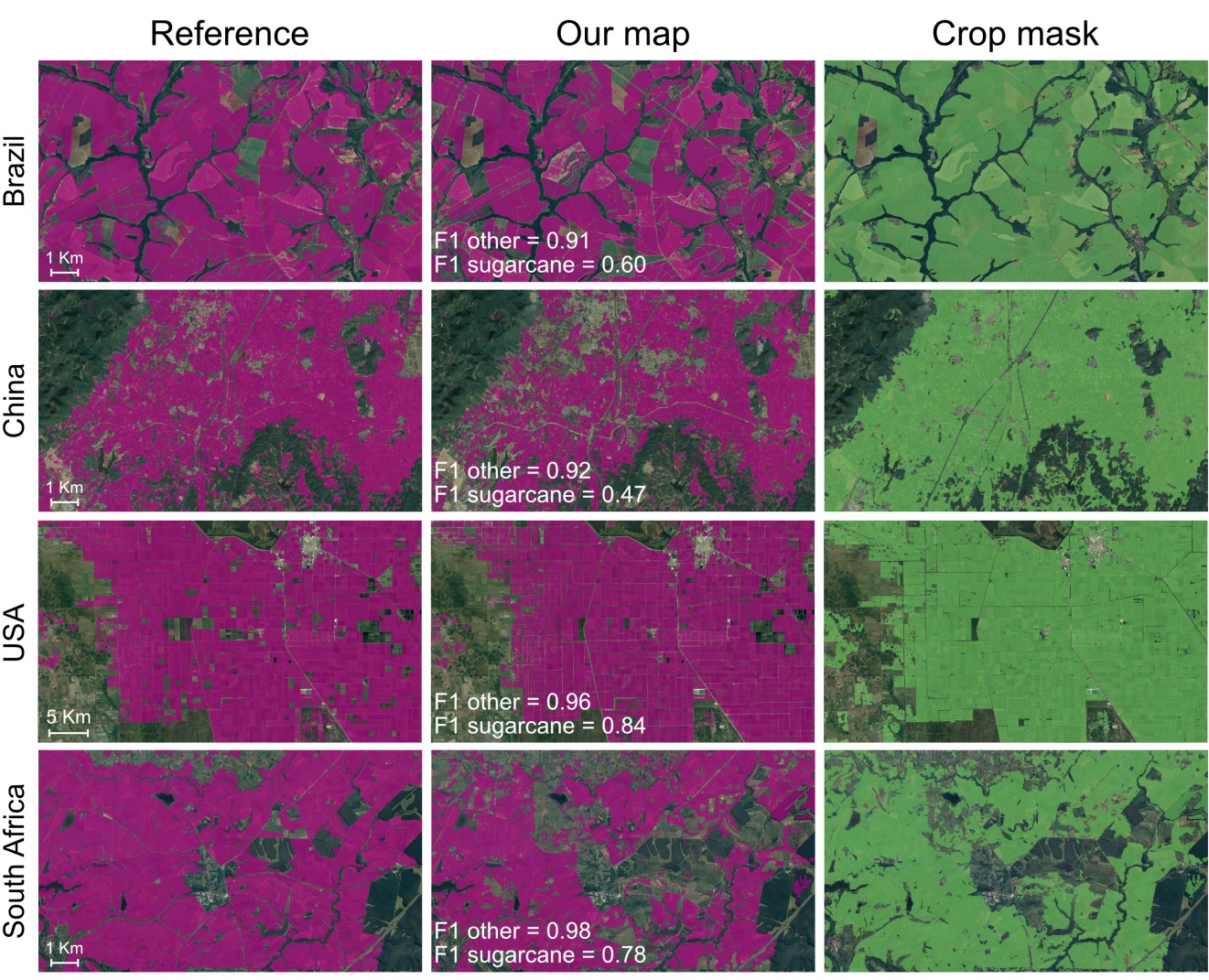

**Figure 6.** Comparison between our sugarcane maps and reference rasters. Maps show zoom-ins of key sugarcane-producing regions in Brazil, China, US and South Africa (© Google Earth Engine). The reported metrics pertain to the entire region covered by the reference maps, not just the illustrated portions. The absence of sugarcane in certain predicted maps for select regions can be attributed in part to the crop mask selection (ESA+GLAD), which omits certain cropped areas (e.g., South Africa).



Conversely, in South Africa, precision and recall are slightly lower at 0.73 and 0.84, respectively, with a portion of labels for
commercial annual crops misclassified as sugarcane.

### 4.3.3 Validation against government statistics

To evaluate the accuracy of our sugarcane maps, we conducted a comparison with government reported statistics on sugarcane
area. We present the results in Figure 7 at the finest available scale provided by the governments. The only exception is
Pakistan, where we group the data at level 2 due to uncertainties/changes of level 3 administrative division borders over time.
Only administrative regions fully covered by our sugarcane maps are included in these results. We find overall good agreement
with government statistics for the main sugarcane-producing areas. Many countries (6) exhibit an $R^2$ of 0.85 or higher (Brazil
0.92, Pakistan 0.85, USA 0.99, Australia 0.9, Guatemala 0.97, Philippines 0.85).
Some exceptions occur in regions where inaccurate crop masks lead to over prediction of sugarcane area. Specifically,
in Yunnan, China, many orchard areas are included in the crop mask, and because these are tall for the entire year tend to
get classified by our model as sugarcane. Moreover, regions predominantly characterized by (irrigated) maize cultivation, as
evident in Sinaloa, Mexico, also tend to be misclassified as sugarcane by our model, presumably because they are growing
maize every year of the study period. Outside of these problematic regions, the model agrees well with official statistics in each
country. The $R^2$ increases from 0.73 to 0.96 when removing Yunnan in China, and from 0.46 to 0.78 when removing Sinaloa
in Mexico.
Moving to the assessment of main sugarcane-producing states within each country, São Paulo emerges as the main con-
tributor to Brazil's sugarcane landscape, accounting for over half of the planted area. Here, our analysis demonstrates robust
agreement between predicted and government-reported sugarcane areas, with an $R^2$ value of 0.94 and a slope of 0.88, based
on 630 administrative units.
In India, Uttar Pradesh (UP) stands as the primary sugarcane producer, followed by Maharashtra and Karnataka, the three
states together contribute approximately 80% of the nation's sugarcane production. Notably, UP exhibits strong agreement
with government-reported data, with an $R^2$ value of 0.95 and a slope of 1.26. Conversely, while Maharashtra and Karnataka
also demonstrate a good agreement, with $R^2$ values of 0.79 and 0.96, respectively, the regression line slopes for both states is
close to 2 (2.2), suggesting that the predicted sugarcane area is more than twice the reported area.
In China, Guangxi has the highest cultivation land and production of sugarcane, accounting for more than 60% of the total
national area. We observe strong agreement with the government-reported area, with $R^2$ value of 0.97 and a slope of 1.
In Colombia, agricultural statistics are reported at the administrative level 1, known as departments. Our map provides full
coverage solely for the Caldas department, a minor sugarcane-producing region, with an estimated area three times smaller that
the reported area. It's worth noting that the primary sugarcane-producing areas, Valle de Cauca and Cauca, are only partially
covered by our maps. Despite this, we observe substantial agreement between the mapped areas and the government-reported
sugarcane area.
In Indonesia, statistical data on sugarcane production is available at provincial level (admin 1). However, only Jawa Barat, a
minor sugarcane-producing province, is fully covered by our sugarcane map, while the main sugarcane-producing provinces,

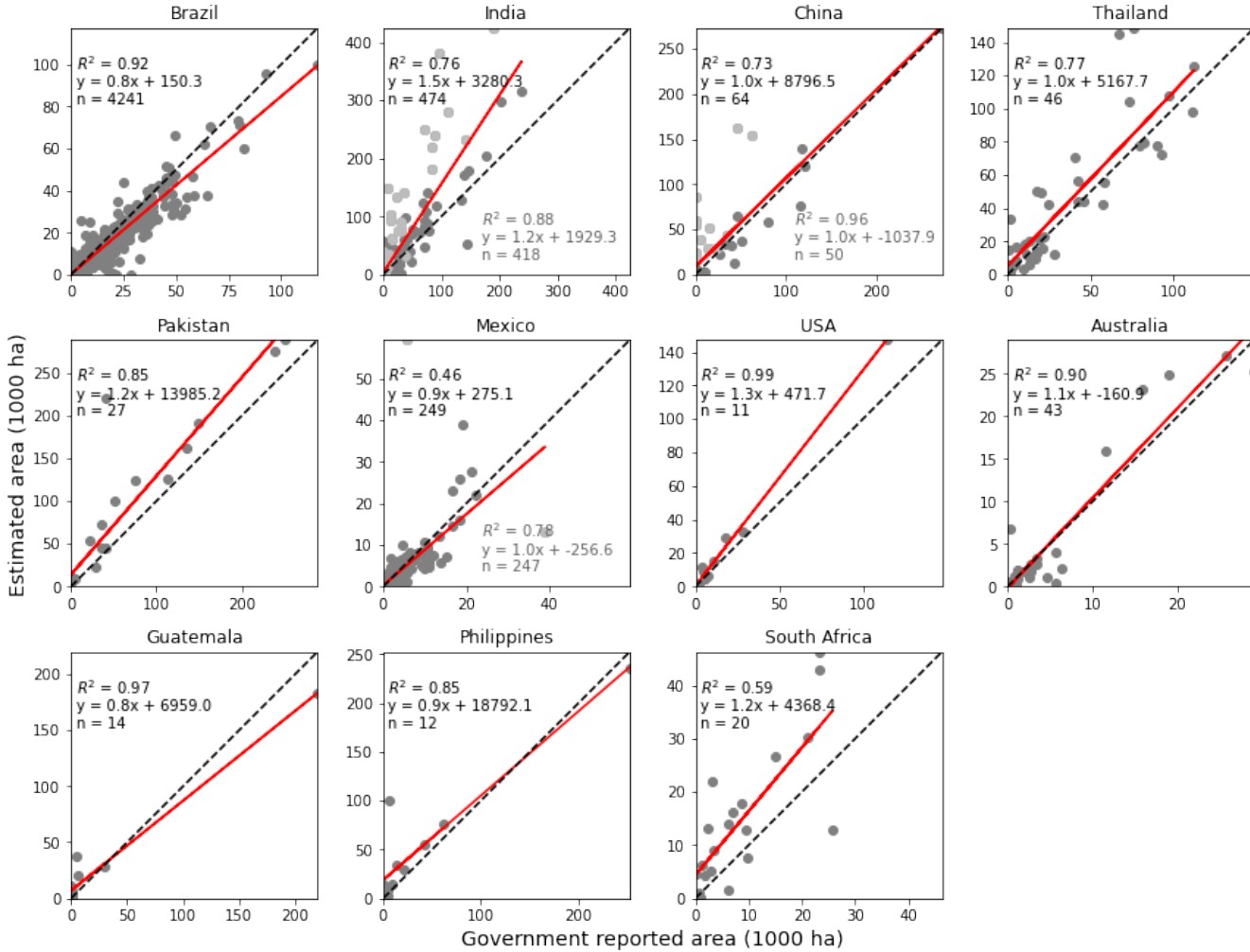

**Figure 7.** Comparison of the sugarcane area in our maps with government statistical data. On the top-left corner are reported the results using all data points. For some countries, on the bottom-right are reported the results removing problematic regions.

such as Lampung and Sumatra Selatan, have partial coverage. Despite this limitation, our analysis reveals agreement between
the reported sugarcane area and the mapped areas in these provinces.
In US, Florida and Louisiana are the main producers, with $R^2$ of 1 and 0.9, respectively. In Australia, Queensland serves as
the primary producer state, demonstrating an $R^2$ value of 0.91.
Regarding South Africa, the available government statistics pertain exclusively to commercial farmers, whereas our analysis
includes all sugarcane fields, encompassing both commercial and smallholder operations. Despite this disparity, we report the
agreement because commercial farmers contribute to over 80% of the total sugarcane production in the country. The lower $R^2$
value may be attributed to the type of reported statistics as well as potential crop mask issues.





## 5 Discussion

### 5.1 Agreement with field data and raster

Synthesizing lessons from point and raster data, we find that GEDI and S2-based sugarcane mapping presents challenges, particularly in regions where tall crops like cassava and maize, especially irrigated maize, coexist with sugarcane. We also observe discrepancies in performance between point and raster data. In Brazil, we noticed lower performance in raster maps (F1 score of 0.6) compared to WorldCereal point data (F1 score of 0.9). This discrepancy could potentially be attributed to the construction of reference rasters, wherein sugarcane is defined as the union of the two most recent years, along with differences in the years considered. In South Africa, it's worth noting that performance against the SANLC field points surpasses that of the South Africa SANLC 2020 map. For field points, the F1 score for sugarcane is 0.9, with precision at 0.97 and recall at 0.84. In contrast, the map exhibits an F1 score of 0.78, with precision and recall values of 0.73 and 0.83, respectively. In the US, where we have high confidence in the CDL maps and reference map years align with our mapping period, we observe good agreement with CDL sugarcane data. However, it is essential to emphasize that reference maps may not be equally reliable, potentially leading to discrepancies in performance evaluation.

### 5.2 Agreement with government statistics

The comparisons with government statistics are complicated by several factors, including the unknown accuracy of official numbers and the fact that they do not necessarily intend to reflect all of cropland area planted with sugarcane. Government data often reflect sugarcane harvested area for a single year, while our mapping captures total-stable sugarcane area over a four-year period. We therefore would expect our numbers to be slightly higher than government numbers, even if both datasets were perfectly accurate. Despite this disparity, we generally observe favorable agreement in most regions.

In India, particularly in Maharashtra and Karnataka, deviations from the 1:1 line are evident, with slope values of 2.3 and 2.2, respectively. Notably, in Maharashtra, the mapped area (2,702,144 ha) exceeds the government-reported area (822,407 ha) by over 230%. However, Plantix data in Maharashtra, which was adjusted for bias in class representation as described in section 2.4.1, revealed a low user's accuracy (50%). This is a warning that commission error associated with the sugarcane class was problematic.

To address this, we employed an error-adjusted estimator of area proposed by Olofsson et al. (2013) to correct the estimated sugarcane area and provide confidence intervals. Taking into account the presence of false positives, consisting of 955 instances among 1,911 sugarcane labels, and false negatives, comprising 529 instances among 13,384 non-sugarcane labels, alongside a proportion of area mapped as sugarcane equal to 0.15, our analysis yielded a revised estimate of sugarcane area of 1,953,625 ha. This revised estimate notably surpasses the reported area by approximately 140%.

The resulting confidence interval, computed using the method suggested by Olofsson et al. (2013), suggests that the sugarcane area estimate could range from 1,873,290 ha to 2,033,960 ha at a 95% confidence level. Despite the wide confidence interval, it is still well above the government-reported area, and the gap is too large to be explained by the difference between total and harvested area. We therefore suggest that the official numbers in Maharashtra are significantly underestimating the





actual sugarcane area. This conclusion is similar to that reached in a previous study in the Upper Bhima Basin within Maha-
rashtra, which concluded that actual sugarcane area may be twice as large as what is indicated in government statistics (Lee
et al., 2022).

## 5.3 Future improvements

A number of future directions could improve the accuracy of our maps. A key dependency in our approach is the use of
existing crop maps that delineate arable cropland from other land uses, including permanent tree crops. Yet we observed in
several regions, most notably in Southern China, that the crop mask often included areas with orchards. Because orchards are
tall throughout the year, removing them from the crop mask is an important need for further improvement. Likewise, in some
regions the crop masks we utilized miss some areas that appear in other sugarcane reference maps (e.g. in South Africa, see
Fig. 6). By improving the accuracy of the crop mask, more precise sugarcane maps can be generated, providing more reliable
information for agricultural planning and management. Implementing subnational thresholds could further refine the accuracy
of our estimations, considering the localized variations in sugarcane cultivation practices. Lastly, integration of other sensor
data, such as Sentinel-1, into our mapping framework could enhance the performance.
In future iterations, extending the grid to encompass more geographical areas could provide a broader perspective on sugar-
cane dynamics. Additionally, extending our maps back in time would allow us to examine changes over time and could offer
valuable insights into temporal trends.

## 6 Conclusions

In this study we have introduced a dataset of sugarcane maps for the top 13 producing countries, covering nearly 90% of global
production, leveraging satellite remote sensing data from GEDI and Sentinel-2 for the years 2019-2022.
Sugarcane cultivation stands as a vital economic activity globally, contributing significantly to food and biofuel production.
With a quarter of the world's ethanol production sourced from sugarcane, countries like Brazil and India are positioned to
substantially increase their ethanol output. However, alongside its economic benefits, sugarcane cultivation presents numerous
social and environmental challenges, including water scarcity, soil pollution, and labor exploitation. Despite its pivotal role in
economies worldwide, reliable information on sugarcane cultivation remains scarce.
Our methodology overcomes limitations of traditional ground-based data collection, offering a scalable approach to mapping
sugarcane canopies globally. Through comparisons with field data, pre-existing maps, and government statistics, we have
demonstrated the accuracy and reliability of our maps.
However, challenges persist, particularly in regions where tall crops like cassava and maize coexist with sugarcane. Addi-
tionally, our approach's dependency on existing crop maps to delineate arable cropland from other land uses presents another
hurdle. These challenges underscore the necessity for ongoing refinement of our mapping techniques.
The final maps should be useful in studying the socio-economic and environmental impacts of sugarcane cultivation and
producing maps of related outcomes such as sugarcane yields.



**Data availability statement**

The final output of our study comprises the frequency of tall mappings for each 10 by 10 m pixel in the combined crop mask (union of ESA, ESARI and GLAD), alongside the sugarcane maps for each country obtained applying the calibration threshold and the crop masks used. Results were provided for each region using the calibration threshold and masking maps using the union of the ESA and GLAD crop masks. However, with the dataset provided, users have the flexibility to use a region-specific crop mask and their own region-specific thresholds if they possess additional insight or calibration data, allowing for customization of the sugarcane mapping process.

The dataset is available on Google Earth Engine at https://code.earthengine.google.com/?asset=projects/lobell-lab/gedi_sugarcane/maps/imgColl_10m_ESAESRIGLAD and for download from Zenodo at https://doi.org/10.5281/zenodo.10871164 (Di Tommaso et al., 2024)

**Author contributions**

SD, SW and DL designed the research. SD developed the model code and performed the analysis. SD prepared the manuscript with contributions from all co-authors.

**Competing interests**

The authors declare that they have no conflict of interest.

**Acknowledgements**

We thank Yuan Wenping for sharing the China sugarcane map, and the Google Earth Engine team for making large-scale computational resources available to researchers.

This work was supported by the NASA Harvest Consortium (NASA Applied Sciences Grant No. 80NSSC17K0652, sub-award 54308-Z6059203 to DBL).



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
