# Peer review of "Mapping sugarcane globally at 10 m resolution using GEDI and Sentinel-2"

_Earth System Science Data, 2024_

## Author Comment (AC1)

**RC1**

Major Comments:

1) The criteria used to classify sugarcane from other crops might not distinguish effectively between bamboo and sugarcane, both of which are tall, perennial members of the grass family. Given the similarities in their growth habits and physical characteristics, there is a risk of misclassification in regions where bamboo is prevalent. It would be helpful to include a sensitivity analysis addressing this potential issue. Adding discriminative remote sensing indices or additional ground truth data to differentiate between bamboo and sugarcane could significantly enhance the classification accuracy.

**AC**: We understand your concern given that bamboo and sugarcane both grow in tropical and subtropical regions of the world and both belong to the Poaceae family. However, while sugarcane is a semi-perennial grass, bamboo is a perennial woody crop. Crop masks from ESA, ESRI and GLAD all exclude perennial woody crops from their definitions, as explained in section 2.3 at line 120-124. However, crops that exhibit similar characteristics in terms of height and growth period to sugarcane might be included in our sugarcane classifications if not appropriately masked out by the crop masks.

We have modified the manuscript to address the possibility that bamboo might be one source of error in our sugarcane maps.
We have added a paragraph to section 4.3.3 "Validation against government statistics" that reads: "It is possible that in India, as in China, a source of our over prediction of sugarcane area is the underlying crop masks, which might include land uses other than arable crops such as permanent tree crops or bamboo."
Section 5.3 'Future improvements' now reads: "A key dependency in our approach is the use of existing crop maps that delineate arable cropland from other land uses, including permanent tree crops and perennial woody crops like bamboo."

We have also modified the abstract to clarify that sugar is unique among cropland species: "Sugarcane was then identified by leveraging the fact that among all non-tree species grown in cropland areas, sugarcane is typically tall for the largest fraction of time."

2) The manuscript used a uniform threshold ("Tall month") across different countries for sugarcane classification, which might not account for regional variations in sugarcane phenology influenced by local climatic conditions and sugarcane varieties. In Figure 3 we see that the shape of the curves of the kappa coefficients in response to the threshold varies considerably from country to country. It may be helpful to explore the diversity of sugarcane cultivated species within different countries and region-specific thresholds to improve classification accuracy and to account for uncertainty in some regions.

**AC**: To account for variations in sugarcane phenology across different countries we use a distinct "tall month" threshold for each country, as reported in Table 3 of the manuscript. We acknowledge that there may also be variation within each country and that's part of future improvement of these maps. For instance, in India, where we have a substantial number of Plantix samples, we observed some differences between the states of Maharashtra and Uttar Pradesh. While the overall threshold for India is 8 months, the optimal thresholds are 9 months for Maharashtra and 7 months for Uttar Pradesh.
Unfortunately, we lack the amount of data necessary to conduct such detailed analysis at a subnational scale. The solution we propose is to provide, along with the sugarcane map produced using our defined threshold, a layer indicating the number of tall months. In this way, if users have more local data, they can create a more accurate local map.

[Figure]

We have added a paragraph to section 4.1, which reads: "Some regions may experience subnational variation in sugarcane cultivation practices. For instance, in India, where we have a substantial number of Plantix samples, we observed some differences between the states of Maharashtra and Uttar Pradesh. While the overall threshold for India is 8 months, the optimal thresholds are 9 months for Maharashtra and 7 months for Uttar Pradesh. Unfortunately, in the current study, we lack the amount of data necessary to conduct such detailed analysis at a subnational scale for all countries."

3) The validation results show a significant discrepancy between the F1 (0.64) and R2 (0.97 with a slope of 1) in Guangxi. A more detailed investigation into these discrepancies is warranted. Otherwise, it is difficult to distinguish whether the match between the sugarcane planting area obtained and the government report is a coincidence or not.

**AC**: That's a valid concern, but we want to note that the 0.64 F1 score in Guangxi refers to validation against a remote sensing based product. The reported performance of the reference sugarcane map used in Guangxi are an overall accuracy of 93.81%, and producer's and user's, of 86.31% and 87.89%, respectively. The low performance could also be attributed to the way we constructed the reference map, which is the union of the two most recent years available, 2019 and 2020. We also note that we have similar low performance in Brazil when comparing

our maps against another raster map, with an F1 score of 0.6. However, in Brazil, we had more than 3,000 samples from WorldCereal, against which we achieved an F1 score of 0.9. We believe that the lower performance we observe in some countries when validating against other remote-sended rasters is due to the additional error introduced by the modeled data, as we suggest in section 5.1 'Agreement with field data and raster'. While in Brazil we have separate field-level samples to support this, in China we have no additional data to confirm it.

We have modified the manuscript in section 5.1 "Agreement with field data and raster" that now reads: "In Brazil, we observed lower performance in raster maps (F1 score of 0.6) compared to WorldCereal point data (F1 score of 0.9). This discrepancy could be attributed to the construction of reference rasters, wherein sugarcane is defined as the union of the two most recent years, along with differences in the years considered. In Guangxi, China, we observed similarly low performance (F1 score of 0.64) when comparing our maps to modeled raster data, despite high agreement with government statistics, which also indicates potential errors in the construction of the raster reference maps."

4) Have the epidemic and climate change had a large impact on sugarcane planting? What specific year's sugarcane extent does map reflect? Or is it the combined acreage for the three years from 2019 to 2022?

**AC**: That's a good question. The map we propose is a combined representation for the years 2019-2022 due to the nature of sugarcane as a semi-perennial crop. As described in the future improvements section, the current study did not attempt to create yearly maps or track changes over time, but that would be an area for future development.

We have modified the last paragraph of section 5.3 "Future improvements" as follows: "Additionally, another interesting direction for future research would be to extend our maps back in time. This would allow us to examine changes over time, observe the impact of climate change on sugarcane plantations in various regions, and provide valuable insights into temporal trends."

Minor Comments:

1) The heading "Area" in Table 1 might change into "Country" to align with Table 2 & 3.

**AC**: Thank you. We have updated Table 1 to use "Country" as header.

2) Line 52, the abbreviation GEDI should appear after the first occurrence of the full name.

**AC**: Updated to: "Global Ecosystem Dynamics Investigation and Sentinel-2 Sensors: Data from the Global Ecosystem Dynamics Investigation (GEDI) and Sentinel-2 (S2)"

3) Line 82, what is the threshold of the cloud probability you used for masking?

AC: Thanks for the suggestion, in the text, we have added the threshold used: "Clouds were filtered out using the S2 Cloud Probability dataset provided by SentinelHub in GEE setting MAX_CLOUD_PROBABILITY to 65"

**RC2**

Dear authors,

How to map sugarcane is vital especially at global scale, this study makes full use of GEDI and Sentinel-2 imagery to generate the sugarcane maps for top 13 producing countries, and achieving >80% agreement.

However, there are several issues in the current manuscript as:

1. The novelty of the method is weak, it has been published in their previous works in 2023. The scope of ESSD aim to be innovative not only in terms of results, but also in terms of methodology.

AC: While our previous work categorized crops broadly as tall or short, this work introduces a novel approach focusing specifically on sugarcane cultivation. This study delves deeper into the unique challenges posed by sugarcane mapping, accounting for its semi-perennial nature and targeting specific countries where sugarcane is a prominent crop, providing detailed insights into its distribution. We provide maps for sugarcane at a scale (global), resolution (10m), and for a recent period(2019-2022) that have not been previously documented. All of these aspects move well beyond the prior work. Also, for this journal the key criteria appears to be whether the datasets produced are useful, new, and robust, not whether the methodology is novel.

2. The method cannot convince me in some key steps.
   How to generate accurate training samples? Authors mentioned that the GEDI can provide the canopy heights, however, the error of GEDI cannot be directly ignored. I think that the quality control in the GEDI data on GEE cannot solve the vertical error. Meanwhile, we also think only GEDI dataset cannot be used to derive high-confidence training samples, for example, the height of maize also reached tall height, so how to distinguish maize and sugarcane. How training samples for other land classes are obtained? How many training samples were used? The quality and size of training samples greatly affected the accuracy of mapping.

AC: The reviewer appears to misunderstand the approach so we have revised the manuscript to clarify. In our previous work on GEDI we demonstrated that the effect of GEDI errors can be minimized through appropriate quality filtering, allowing us to distinguish tall crops (such as

maize and sugarcane) from shorter crops. Filtering includes the use of GEDI quality and degrade flags, our model confidence score, view angle and slope information. All of these are mentioned in section 3.3 "GEDI data processing". It's important to note that we don't rely solely on canopy height information for distinguishing tall from short crops; we utilize the full waveform, represented by RH metrics as mentioned in section 2.1.

Additionally, the issue of noisy labels is mitigated by the abundance of samples within each grid cell. Thanks to a pooled model that combines four years of GEDI orbits, the majority of grid cells have a sufficient number of training labels to train a robust S2 model capable of distinguishing short from tall vegetation. Most of the 2x2 degree grid cells have more than 9.4k samples used for training the S2 model, with the 5th percentile and 95th percentile of the number of training labels being 768 and 69k samples, respectively.

We appreciate this point and have added the information on the number of training samples in the manuscript in section 3.4 "S2 model training and classification", reading: "Most of the grid cells have more than 9,400 GEDI training labels used for training the S2 model, with the 5th percentile and 95th percentile of the number of training labels being 768 and 69,000 samples, respectively."

The reviewer is correct that the height of other crops, like maize, could be similar to sugarcane. We are therefore not using GEDI by itself to distinguish between maize and sugarcane. The novelty we introduced to distinguish them is based on the length of the growing season. Sugarcane is unique in this respect as it remains tall in the field for a longer period than maize. The calibration of the "tall month-threshold" explained in section 3.5 helps in this task.

We have modified section 3.5 "Calibration/Sugarcane identification" to better explain this point: "To distinguish sugarcane from other tall crops, such as maize, we computed the frequency of tall predictions for each pixel across the 48 monthly predictions. Pixels were classified as sugarcane if the frequency of tall predictions exceeded a certain threshold, based on the principle that sugarcane remains tall for longer periods of time compared to annual crops like maize."

> Section 3.4, you reduce spatial artifacts during the mosaicking of adjacent cells by creating predictions for pixels in a 0.2o, it doesn't convince me either. Actually, you trained the classification models in each 2o×2o tile, so the spatial artifacts were caused by the difference in trained classification models.

AC: We don't observe significant imprints from scene borders or major artifacts due to the use of separate S2 models for each grid cell. This suggests that models from adjacent grid cells are robust. The buffering and subsequent mosaicking on a monthly basis contribute to creating an even smoother map.

We have modified the manuscript in section 3.4 that now reads: "In order to reduce spatial artifacts that may arise during the process of mosaicking adjacent cells, we implemented a strategy where we generated predictions for pixels within a 0.2 degree buffer around each cell.

This buffer ensured that neighboring cells had overlapping coverage. Subsequently, on a monthly basis, we performed the mosaicking process, selecting for the overlapping regions the predictions from the cell with the higher GEDI-S2 kappa score. This ensured that the final mosaic maintained the highest possible accuracy, enabling a smoother transition between adjacent regions."

We have also added a paragraph commenting on the visual quality of the final maps in section 4.2 which now reads: "The sugarcane maps for the main producing countries, obtained applying the previously identified calibration threshold across the 48 monthly predictions, are shown in Figure 4. These maps exhibit high quality, with no significant imprints from scene borders or major artifacts, despite using separate S2 models for each grid cell. This indicates that the models from adjacent grid cells are robust. Additionally, the buffering and subsequent monthly mosaicking processes contribute to creating an even smoother and more cohesive map."

How to use the crop mask in ESA, ESRI and GLAD data is also unclear.

AC: We produce sugarcane maps for the union of all three crop masks. Validation results are instead provided masking our sugarcane maps using the union of the ESA and GLAD crop masks. This is described and justified in the main text at L321-324. We acknowledge that existing crop masks aren't perfect, and each one could be better suited to a particular region. alongside our sugarcane maps, we provide the crop mask layers as three separate bands so that users can choose whichever crop mask they find most suitable for their specific study region. We have added a GEE example script in Zenodo to help the user working with the dataset, see the next comment.
This is the link to the GEE script we are including in Zenodo:

https://code.earthengine.google.com/d4d43daf0b059a553f2ed75f2cb1cf1c?asset=projects%2Flobell-lab%2Fgedi_sugarcane%2Fmaps%2FimgColl_10m_ESAESRIGLAD

Results

1. The classification maps are generated in each 2o×2o tile, and the relationships between tall months and kappa score in Figure 3 are analyzed at national scale. So how to determine the thresholds for tiles that span multiple countries.

AC: Thresholds are applied at country-scale. Final maps are generated per country, as mentioned at L513, meaning that a tile overlapping two countries has the country-specific threshold applied to pixels belonging to that country. In Zenodo, we have added the following link to a Google Earth Engine script to help the users visualize the sugarcane maps by country, and apply a selected crop mask.
https://code.earthengine.google.com/d4d43daf0b059a553f2ed75f2cb1cf1c?asset=projects%2Flobell-lab%2Fgedi_sugarcane%2Fmaps%2FimgColl_10m_ESAESRIGLAD

We have modified the Data availability statement as follows: "The dataset can be accessed on Google Earth Engine at https://code.earthengine.google.com/?asset=projects/lobell-lab/gedi_sugarcane/maps/imgColl_ 10m_ESAESRIGLAD. Additionally, users can find the dataset on Zenodo, at https://doi.org/10.5281/zenodo.10871164. In the Zenodo repository, users will also find a link to a GEE script for visualizing and masking the sugarcane maps by country."

> More descriptions about the Section 4.3.2 should be greatly strengthen, for example, why China achieved the lower F1 score of 0.47?

**AC**: The low China F1 score of 0.47 refers to the validation result against raster data for four nations. Unfortunately, no sugarcane field data is available for China to our knowledge. To perform point-level validation, we opted to use the only available sugarcane map, a remote-sensed product, which may have its own accuracy issues and may not exactly match the years in our study. We also acknowledge that Yunnan, in particular, presents strong errors due to an inaccurate crop mask that includes many orchards, as mentioned at L406.

We have modified section 5.1 that now reads: "In Brazil, we observed lower performance in raster maps (F1 score of 0.6) compared to WorldCereal point data (F1 score of 0.9). This discrepancy could be attributed to the construction of reference rasters, wherein sugarcane is defined as the union of the two most recent years, along with differences in the years considered. In Guangxi, China, we observed similarly low performance (F1 score of 0.64) when comparing our maps to modeled raster data, despite high agreement with government statistics, which also indicates potential errors in the construction of the raster reference maps."

**RC3**

This study introduced an innovative algorithm for sugarcane mapping using GEDI and Sentinel-2 data and published the resulting mapping dataset. The GEDI data was used to derive the tall and short crops to train Sentinel-2 optical data to derive wall-to-wall monthly short and tall crop maps, which is then thresholded to derive sugarcane maps. The thresholds are defined from training samples collected from different formats and sources. The results are reasonable over most countries except a few countries where the sugarcane is mixed with corn or cassava. This reviewer evaluated only the manuscript, not the maps.

I have a few comments on clarity on the paper.

Line 17, What is the full name of OECD?

**AC**: Thanks for pointing it out, OECD stands for Organization for Economic Co-operation and Development.
We have now added the full name in the manuscript, it now reads: "For example, the Organisation for Economic Co-operation and Development (OECD) and the Food and Agricultural Organization (FAO) project that ethanol demand over the next decade will increase

by 37% in Brazil and 107% in India (OECD et al., 2023), both countries where sugarcane is the primary feedstock"

Lines 26-28, the sentence does not make sense to me, please rephrase "sugar receives commodity-specific transfers of more than 20% of farm receipts globally, higher than any other food commodity"

**AC**: Thanks for pointing this out. We have replaced the phrase 'commodity-specific transfers' with the more common phrase "subsidies" for clarity. It now reads: "According to recent OECD estimates, sugar subsidies represent more than 20% of farm receipts globally, higher than any other food commodity"

Line 73, 1.51 rad? Can you use unit degrees?

**AC**: We have added the corresponding angle in degrees to the manuscript for easier interpretation by the reader. We retained the values in radians because the GEDI local beam elevation property is originally expressed in radians.
We have modified the line to read: "This information was used to filter out GEDI shots with a view angle below 1.51 rad, approximately 86.5 degrees, to avoid classification errors"

It is good to see the authors use various sources of samples.

**AC**: We truly value your acknowledgment of our efforts to incorporate all available sources in our work.

Line 155-156, rephrase the sentence

**AC**: Thanks for catching that. We have corrected the sentence as follows: "Labels include sugarcane, cassava, maize, rice, and an "other" crop class. To ensure the labels were representative of the landscape, they were sampled in alignment with government-reported crop areas."

What is the definition of tall and short crops?

**AC**: At line 252, we refer the reader to our previous paper Di Tommaso et al. (2023), where we train the GEDI model to distinguish tall vs short. This model is trained on high-accuracy crop type labels from the three regions. The tall class is represented by maize samples, and the short class is a mix of mostly soybeans, rice and spring barley. A detailed description of type and number of labels is provided in Figure A1 of Di Tommaso et al. (2021).

We have modified the manuscript at L254 to add this information: "All GEDI shots from April 2019 to December 2022 over cropland pixels, passing over these 2 degree x 2 degree grid cells, were classified as either short, tall, or tree by a GEDI model trained in Di Tommaso et al. (2023). This model is trained on high-accuracy crop type labels from the three regions. The tall class is represented by maize samples, and the short class is a mix of mostly soybeans, rice and spring barley labels. Each classification was accompanied by a confidence value."

Section 4.2 needs more comments.

AC: We have added a paragraph commenting on the visual quality of the final maps. The 4.2 section now reads: "The sugarcane maps for the main producing countries, obtained applying the previously identified calibration threshold across the 48 monthly predictions, are shown in Figure 4. These maps exhibit high quality, with no significant imprints from scene borders or major artifacts, despite using separate S2 models for each grid cell. This indicates that the models from adjacent grid cells are robust. Additionally, the buffering and subsequent monthly mosaicking processes contribute to creating an even smoother and more cohesive map."

The authors in fact used a decision fusion method where monthly classified tall short crops are fused. An alternative way is to fuse the time series by using deep learning models. This is feasible since the Transformer can classify raw irregular time series data, as demonstrated by several studies. Can the authors discuss the possible other fusion methods? In particular consider that the authors are experts on deep learning applications.

AC: thanks, this is an interesting idea and one that could be explored in future work. Our goal here was to use methods that could easily scale computationally in the Google Earth Engine platform in order to produce global-scale maps. But as computations limits change, it may become more feasible to run deep learning models at these scales. We have added a sentence to section 5.3: "Integration of other sensor data, such as Sentinel-1, as well as other approaches to summarizing time series than the harmonic regressions used here, could enhance model performance."

**RC4**

While the article is well-written and presents a significant advancement in agricultural mapping, it does not clearly explain how sugarcane is differentiated from other tall crops globally. For instance, bamboo can visually resemble sugarcane, and it is not clear how the methodology distinguishes between such similar tall crops on a global scale. Clarification on the criteria and processes used to ensure accurate differentiation would enhance the robustness of the study's findings.

Aside from this concern, the article is comprehensive, methodologically sound, and should be accepted for publication.

**AC**: We appreciate your feedback and understand your concern regarding the differentiation of sugarcane from other tall crops globally. In our study, we rely on global crop masks to identify arable land and distinguish sugarcane from other crops. Specifically, we utilize crop masks from sources such as ESA, ESRI, and GLAD. As detailed in section 2.3 (lines 120-124), these crop masks explicitly exclude tree crops and perennial woody crops, including bamboo.
We acknowledge that while these crop masks are designed to accurately delineate arable crops, there may be regional imperfections. In some cases, crops with similar physical characteristics to sugarcane, such as height and growth period, might be erroneously classified as sugarcane if not properly excluded by the crop masks.

We are aware of these limitations and we have modified the manuscript to address the possibility that bamboo might be one source of error in our sugarcane maps:
In section 4.3.3 "Validation against government statistics" line 398 now reads: "It is possible that in India, as in China, a source of our over prediction of sugarcane area is the underlying crop masks, which might include land uses other than arable crops such as permanent tree crops or bamboo."
In section 5.3 "Future improvements" line 453 now reads: "A key dependency in our approach is the use of existing crop maps that delineate arable cropland from other land uses, including permanent tree crops and perennial woody crops like bamboo. "

We have also modified the abstract to clarify that sugar is unique among cropland species:
"Sugarcane was then identified by leveraging the fact that among all non-tree species grown in cropland areas, sugarcane is typically tall for the largest fraction of time."

---

## Author Response (AR2)

**Dear Editor,**

**Thank you for the reviewers' thorough examination of our manuscript and data product.**
**We have carefully reviewed the locations identified by RC1. We found that most of the locations flagged as errors in our classification map are actually filtered out when applying the ESA or GLAD crop mask, as recommended in the paper. Additionally, one location is outside our study area.**
**Our responses to the reviewers' comments are highlighted in blue in the attached document. We appreciate all the feedback received.**
**Thank you once again.**
**Best regards,**

**Stefania Di Tommaso (on behalf of all authors)**

**RC1**

It is great to see the detailed explanation and revisions from the authors. However, more questions arise following their updates.

Upon examining via the Google Earth Engine, I observed significant variability in the quality of the sugarcane map among different countries. The map performs quite well in American countries such as Brazil, the USA, and Mexico – excellent job! However, there are considerable uncertainties in the product when it comes to Asian countries such as China, India, and Indonesia.

In China, many forests are mistakenly mapped as sugarcane. In India, it appears that many sugarcane fields are omitted, and some forests are incorrectly classified as sugarcane. Similarly, in Indonesia, many palm trees are classified as sugarcane.

Here are some specific regions where I randomly encountered these issues using high-resolution satellite images on Google Maps:

China: [112.6519455701914, 31.93629809997078]; [108.09442950178136, 34.20204491117551]
India: [79.50481993956686, 28.244005880827807]
Indonesia: [104.9733662708124, -3.0038818470766]; [104.39254177115578, -2.441290301781308]
It seems that simply using the threshold of vegetation height and land cover classification maps can't effectively achieve sugarcane mapping across multiple countries. The mapping accuracy is greatly affected by the quality of the land cover maps especially in countries with various crop/forest types. Additionally, I am uncertain whether palm trees are categorized under "crop" in some land cover maps, which might explain why large areas of palm trees in Indonesia are classified as sugarcane.

To address these issues, I suggest introducing a quality grading system for products from different countries. Such labeling would help users make more informed decisions about the data's reliability and suitability for their specific needs.

All in all, this manuscript is well written, but the quality of the products appears to differ significantly across countries and regions. Therefore, I believe that the current final product is not yet mature enough for use on a global scale. I think it would be suitable for publication if the authors can address the issues mentioned above.

We appreciate the thorough review of our paper and data product.

We would like to emphasize that the provided sugarcane map needs to be masked using an appropriate crop mask, as suggested in the paper in the Data availability section. Along with the sugarcane layer, three additional bands are included in the data, representing crop masks from ESA, ESRI, and GLAD. We encourage users to apply one of these crop masks, a combination of them, or an external crop mask that is best suited for the specific study area. The Google Earth Engine script provided included examples of applying crop masks.

Most of the errors identified by the reviewer in our maps are filtered out when using the ESA or GLAD crop masks. We note in the paper in sec. 2.3 that the ESRI crop mask in most regions tends to overclassify crops, including orchards and trees. This is why in the paper, for validation against government statistics, we use the union of the ESA and GLAD crop masks, which is our preferred choice as explained in sec. 4. Additionally, one of the locations in China mentioned by the reviewer is outside our study region.

Brazil is one of the regions where the ESRI mask helps capture some cropland missed by the ESA mask in the northeastern part, as illustrated in figure 2. This is why we are providing sugarcane predictions for the ESRI crop mask pixels as well.

We have modified the script to display the ESA+GLAD crop mask as the default (instead of the ESA+ESRI+GLAD combination) and added a note in the GEE script to remind users to appropriately select a crop mask. The note reads "USAGE: Users must first choose a country and then select the most appropriate crop mask from the ones provided for that specific region. If none of the provided crop masks are suitable, users can use an external crop mask instead." You can access the updated script here: https://code.earthengine.google.com/545a87ce9bc29f2b5ad180955d974f8c?asset=projects%2fl Bell-lab%2Fgedi_sugarcane%2 Maps%2FimgColl_10m_ESAESRIGLAD

On the Zenodo page, the note reads: "USAGE: Users must mask the provided sugarcane map with the most appropriate crop mask from the ones provided. If none of the provided crop masks are suitable, users can use an external crop mask instead."

**RC2**

Accepted as is.

Thank you for appreciating the originality and data quality of our work.

---

## Author Response (AR3)

Dear Editor,

We have added a note in the Zenodo repository to address the reviewer's comment regarding the limitations faced during the validation process in some regions.

Our response to the reviewers' comment is highlighted in blue in the attached document. We appreciate all the feedback provided.

Thank you once again.
Best regards,

Stefania Di Tommaso (on behalf of all authors)

**RC1**

I would suggest that sugarcane mapping without raster or field data validation not be put into a formal dataset, but rather into an immature dataset as supplementary data (For example, the data for Indonesia). Government statistics alone can not guarantee the accurate mapping. No more comments beyond that. Good luck!

Thank you for your feedback. We have addressed your concern by adding the following note to our Zenodo repository:
"Validation results for the sugarcane maps are detailed in Section 4.3 of the paper. For Indonesia and Guatemala, no field-level data or raster datasets were available for validation of our sugarcane maps."